# Advanced Activities of Daily Living in Community-Dwelling Older Adults: A Cross-Sectional Study of the Mexican Health and Aging Study (MHAS 2018)

**DOI:** 10.3390/healthcare11142107

**Published:** 2023-07-24

**Authors:** Martha A. Sánchez-Rodríguez, Mariano Zacarías-Flores, Elsa Correa-Muñoz, Víctor Manuel Mendoza-Núñez

**Affiliations:** 1Research Unit on Gerontology, FES Zaragoza, National Autonomous University of Mexico, Mexico City 09230, Mexico; elcomm_unam@yahoo.com.mx (E.C.-M.); mendovic@unam.mx (V.M.M.-N.); 2Division of Obstetrics and Gynecology, Hospital Gustavo Baz Prada, Institute of Health of the State of Mexico, Nezahualcóyotl 57300, State of Mexico, Mexico; mzacariasf@yahoo.com

**Keywords:** advanced activities of daily living, AADL, MHAS, older adults, functional capacity, healthy aging

## Abstract

Background: The advanced activities of daily living (AADLs) in old age is a key indicator of the mobility domain for the intrinsic capacity of older adults living in the community; for this reason, it is relevant to know the prevalence and risk factors related to performing fewer AADLs in different populations. Aim: To determine the prevalence and factors associated with the ability to perform AADLs in older adults reported in the Mexican Study of Health and Aging (MSHA 2018). Methods: A secondary cross-sectional analysis of the MSHA 2018 data was carried out, including a convenience sample of 6474 subjects ≥ 60 years of age, for both sexes, without cognitive deficits. Nine questions related to AADLs were selected from the database. Multiple logistic regression analysis was performed to determine factors associated with <3 AADLs, including sociodemographic, lifestyle, and health status factors. Results: The prevalence of the ability to perform <3 AADLs was 63%. Age is the most important risk factor for <3 AADLs, which increases by the decade, followed by sedentary lifestyle (OR = 2.15, 95% CI: 1.91–2.42, *p* < 0.0001). Conclusions. Our findings suggest that age, schooling, urban residence, sedentary lifestyle, and comorbidity are the main risk factors for <3 AADLs in older Mexican adults.

## 1. Introduction

Old age is the last stage of the human life cycle, in which the organism falls into a gradual functional decline, with the loss of physiological integrity, increasing the probability of death [1,2]. This process is complex, and it depends on several factors such as the chronological age, genetic load, environmental factors, and lifestyle, [1,2,3]; thus, the aging is an individualized and heterogeneous process.

In this regard, since 1987, Rowe and Khan described two ways of aging, usual, or successful. In both types of aging, the elderly have minimum physiological loss and retain their independence [4], unlike that of the weak elderly, who have poor health and a high risk of disability, hospitalizations, and death, and are referred to as frail [5]. This proposal has opened a wide field in gerontological research.

It is important to highlight that within the heterogeneity of aging, not all older adults are frail, because a high percentage of older adults maintain their intrinsic capacity up to 74 years of age, maintaining their physical and cognitive functional capacity even after facing a stressor, because there is both physical and psychological resilience. Therefore, disability is not necessarily an inherent consequence of aging, since the percentage of independent adults older than 85 years is increasing [5,6]. 

In Mexico, more than 25% of these persons have some physical limitation to carry out basic activities such as walking, performing personal hygiene, eating, dressing, or going to the bathroom; 5% of elders cannot perform ≥ 2 of those activities; on the contrary, more than 80% of these old adults are totally independent, and they can participate in active and healthy aging programs in the community [7,8,9]. One of the key elements for successful aging (SE), the maintenance of physical and cognitive functions, is the absence of chronic noncommunicable disease (NCD). Thus, mobility is essential to take into consideration, in addition to social engagement and productive activities [10,11,12]. This approach is very limited, since it excludes a high percentage of older people who could be classified as healthy, even if they do not meet the SE criteria.

Therefore, people with some physical limitations can maintain their mobility with the help of a device; thus, the World Health Organization (WHO) called functional capacity the interaction between the person and their environment; it defines healthy aging (HA) as “the process of promoting and maintaining the functional capacity that allows well-being at the old age” [13].

A way to assess the functional capacity is determining the difficulties in performing the activities of daily living (ADLs), which are stratified according to complexity level as basic, instrumental, and advanced. The basic ADLs (BADLs) are fundamental activities to maintain autonomy and independence for basic physical needs. The instrumental ADLs (IADLs) are the necessary activities to maintain an independent life in the community; and the advanced-ADLs (AADLs) are complex activities of leisure and self-development aimed at preserving cognition. These ADLs are interdependent, in such a way that an older adult that performs the AADLs without difficulty can carry out IADLs and BADLs. When cognitive function declines, gradual problems develop in performing AADLs; later, the execution of IADLs is affected, and finally, the incapacity to perform the BADLs [14]. An evaluation of the AADLs needs to consider motivation-dependent activities such as recreation, volunteering, and educational activities, as well as social participation in the community [15]; however, we must consider the sensorimotor changes and the decrease in speed, proprioception, and reaction capacity that are relative to aging. If the physical and cognitive stimulation is maintained through moderate supervised physical exercise, the evolution of age-related changes can be modified [16,17].

In this sense, the tools used in studies of the functional status of the elderly mainly assess the relationship between BADLs and IADLs with different health statuses [18]. Furthermore, as AADLs are the functional activities that are first lost with cognitive decline, some studies have focused on disability, analyzing the cognitive decline of aging, [14,15,19], without knowing the functional capacity to perform AADLs of community-dwelling older adults with cognitive functioning. 

Several studies have used the Mexican Health and Aging Study (MHAS) for evaluating ADLs with the 2001, 2003, 2012, and 2015 waves; however, their evaluations do not focus on the specific measurements of AADLs and disabilities. In this sense, Gutiérrez-Robledo et al. (2021) carried out a cross-sectional study of the 2015 MHAS, in which they evaluated intrinsic capacity, limiting the mobility measurement to “difficulty walking several blocks” and “difficulty climbing several flights of stairs without resting” [8]. Likewise, Sánchez-Garrido et al. (2021) carried out a study in waves 2012 and 2015 of the MHAS, in which they evaluated the relationship between the social vulnerability index and mortality and disability, evaluating the latter through the measurement of basic activities of daily living [20]. Likewise, Gutiérrez et al. (2020) carried out a study of waves 2012 and 2015 of the MHAS, with the aim of evaluating the relationship between the use of time in some advanced activities of daily living with depressive symptoms, in which it was reported that the volunteer and community activities domain was associated with lower odds of depressive symptoms for women (OR: 0.72, 95% CI: 0.58–0.89) [21]. On the other hand, Arroyo-Quiroz et al. (2020) carried out a longitudinal study of the 2001, 2003, 2012, and 2015 MHAS waves, in order to identify the factors associated with healthy aging in septuagenarians and nonagenarians, in which AADLs were not considered, in the evaluation of their functional capacity [22]. In this framework, our study highlights the importance of AADLs as a key indicator for healthy aging in the community. Therefore, the aim of this research was to determine the prevalence and factors associated with the ability to perform the AADLs in older adults, without limitations on cognition, from the 2018 wave of the MHAS.

## 2. Materials and Methods

### 2.1. Study Description

We carried out a secondary cross-sectional analysis from 2018 wave data of the MHAS [23]. This is a national longitudinal study of community-dwelling Mexican adults aged 50 years and older, who answered questionnaires that assess different topics: demographics, health in multiple domains, socioeconomic conditions, income, and psychosocial aspects, among others. The aim of the 2018 MHAS was to obtain information about the aging process in the population of 50 years and older, their health, disability conditions, socioeconomic status, work activities, and other characteristics, in order to evaluate the impact of disease and disability to perform daily activities [24,25]. Data are available from the webpage http://www.mhasweb.org/ (accessed on 2 October 2020), after registration.

Adults aged 60 years and over, from both sexes, without cognitive deficits, who answered at least eight of the nine questions analyzed, were included in the study. The cognition was determined from whether the elders did not know the complete date, and could not recall at least three words in an immediate word recall test, as was previously suggested in [8].

### 2.2. Advanced Activities of Daily Living

A questionnaire has been proposed and validated to evaluate the AADLs, which consists of 13 items divided into three domains: leisure, social, and productive activities [15,26]. Another study proposed 12 activities to assess the AADLs, some of which are like the previous questionnaire, and added other activities [27]. In the MHAS, only nine of those activities were evaluated; thus, we chose the questions representing the physical/leisure, social, and productive domains, corresponding to three questions each. Each answer was scored 0 (no) or 1 (yes) if the older adult carried out the activity, and the total of activities performed was computed. The MHAS questions included in this research are shown in Table 1.

To determine the reliability and consistency of the proposal construct, we chose 50 participants randomly from the subsample of the older adults that met the inclusion criteria. This selection was performed with the SPSS tool “select a random sample” (V.25.0 IBM SPSS Statistics, Armonk, NY, USA). The Kuder-Richardson test (KR-20) was performed as a reliability test, to determine consistency among the questions; a sensitivity test was obtained by removing a question and recalculating the KR-20; thus, this process was repeated nine times. All of the calculations were processed with Microsoft Excel 365. As cut-off < 3 AADL was used to classify the older adults with performance deficits; this cut-off value was obtained from the median of the 50 data.

### 2.3. Sociodemographic, Lifestyle, and Health Conditions Data

The sociodemographic variables age; sex; locality size (rural < 2500 inhabitants and urban > 2500 inhabitants) [28]; educational level (illiterate, elementary (6 years of schooling); middle school (3 years), high school (3 years), and college or higher (schooling in university)); and marital status (married/in a civil union, never married, divorced/separated; widowed) were obtained. As lifestyle variables were included smoking (currently smoke cigarettes); alcohol intake (currently drinks alcohol); and being sedentary (no exercise or no hard physical work ≥ 3 times per week). The chronic NCD data (hypertension, diabetes, cancer, heart attack, stroke, lung disease, and arthritis) were categorized as 0, 1, 2, and ≥3 diseases, and were included as health conditions. A dummy dichotomous variable was created as health/one NCD = 0 and ≥2 diseases = 1.

### 2.4. Data Analysis

The analysis was carried out with the older adults who met the inclusion criteria, without the 50 persons selected for the reliability and consistency analysis (Figure 1). The frequencies, percentages, and 95% confidence intervals (95% CI) were calculated from the categorical data, and the means and standard deviations from the quantitative data, which were compared with the chi-square and independent *t*-tests, respectively. The analysis was performed with all of the data, and stratified by sex.

The age was categorized by decades until the centenarian decade. The one-way ANOVA with Tukey’s post hoc test was used to compare ≥ 3 groups. 

Since the capacity to execute AADLs depends on several variables such as age, schooling, some lifestyle factors, and health status, multiple logistic regression analysis was carried out to determine the factors associated with performing < 3 AADLs. The models included all sociodemographic, lifestyle, and health condition factors. The age group, educational level, and marital status were included in the models as ordinal variables; the other variables were dichotomous. Models stratified by sex were also run using the same variables. The 60–69-year age group, rural residence, married/in a civil union, with college or higher education, and healthy or with one NCD, were considered as a control group, as well as being male.

A two-tailed *p*-value lower than 0.05 was considered statistically significant. The data were processed using the software package SPSS V.25.0 (IBM SPSS Statistics, Armonk, NY, USA).

## 3. Results

### 3.1. Characteristics of the Analyzed Data

In the MHAS 2018 wave, 18,249 adults were surveyed [24], in which 9327 individuals with complete data were 60 years and older. Separated by age group, 4135 (44.3%) elders were in the 60–69-year age group, the most prevalent, followed by the 70–79-year age group with 3659 (39.2%); only 6 participants were centenarians. Most of the older adults had a partner; 5883 (63%) had at least one chronic disease; 2425 (26%) had mild cognitive impairment, and 373 (4%) had severe cognitive impairment; and the highest ratio of interviewees was women (5219, 56%). Alcohol intake was a more frequent lifestyle habit in men (1496, 36%), and 2717 (29%) of the total number of participants acknowledged having physical activity (Table 2). A final subsample of 6476 (70%) older adults with appropriate cognitive functioning was used for the analysis (Figure 1).

### 3.2. Reliability and Consistency of the AADLs

The reliability of the questions added as AADLs was almost 60%. The most inconsistency in the construct was when the question “participate in any volunteer work for the community” was removed (KD-20 = 0.466; and when “have a primary paid job” was removed, there is the best consistency (KR-20 = 0.611). This may be because the difference between “doing” or “not doing” the activity was small (36%) compared to the other activities that showed differences between 60 and 80%. Therefore, the contribution of the question “have a primary paid job” to the construct was lower than other variables. All of the results are statistically significant (*p <* 0.0001) (Table 3).

### 3.3. Prevalence of AADLs

Thirty-seven percent of the older adults without cognitive impairment could perform three or more AADLs. The older adults mainly attended religious services (76%) and talked on the phone, used the Internet, etc., to communicate with their relatives and friends (75%); therefore, the social domain was more frequent. In the productive domain, 2005 (31%) of the respondents still had a primary paid job; on the other hand, most elders did not do activities from the physical/leisure domain; consequently, the older adults carrying three or more AADLs were 2363 (37%). The older women did more frequent social activities, while the men did more productive activities, but more men performed ≥ 3 AADLs compared with the women (40% vs. 32%, *p* < 0.0001) (Table 4).

### 3.4. AADLs and Sociodemographic, Health Conditions, and Lifestyle

Between the ages of 60 and 69 years, the interviewees performed more AADLs; the possibility of doing them decreased as age increased in both sexes, although there was a difference by sex, since men aged between 60 and 79 years performed more AADLs than women of the same age (2.67 ± 1.22 vs. 2.43 ± 1.12, *p* < 0.0001). The rural inhabitants carried out more AADLs than the urban inhabitants (2.38 ± 1.24 vs. 2.22 ± 1.13, *p* < 0.0001), and were mostly men.

Education seemed to be a determining factor in the execution of AADLs, since illiterates and those who had only primary studies performed fewer AADLs than those with higher educational levels in both sexes, although in women it was more evident. 

Concerning the lifestyle variables, the older adults who smoked and drank alcohol performed more AADLs, mostly men; and the subjects who had physical activity also did more advanced activities. Finally, the healthy older adults performed more AADLs than their counterparts who had 2 or >3 NCDs (Table 5).

### 3.5. Factors Associated with the Performance of <3 AADLs

Controlling different factors, the age is the most important risk factor for the execution of <3 AADLs. The risk increases by decades, being almost eight times higher in the group of older adults ≥90 years than 60–69-year age group (OR = 8.75, 95% CI: 4.77–16.03, *p <* 0.0001). The next important risk factor is sedentary lifestyle, which increases the risk of performance of ≥ 3 AADLs in 115% of the elderly population (OR = 2.15, 95% CI: 1.91–2.42, *p <* 0.0001). Other risk factors are being illiterate and elementary educational level, urban residence, and having two or more chronic diseases, as well as being female. Paradoxically, alcohol intake is related to executing three or more AADLs.

Separating by sex, the risk factors are different. Age and educational level are risk factors to not being able to perform three or more AADLs, which are higher in women than men. Urban residence and having two or more NCDs are not risk factors in women, contrary to men, in which both factors are statistically significant (OR = 1.79, 95% CI: 1.49–2.15, and OR = 1.71, 95% CI: 1.43–2.06, *p* < 0.0001, respectively). Sedentary lifestyle and alcohol intake are the same in both sexes (Table 6).

## 4. Discussion

One way of establishing the functional capacity of older adults is by evaluating the AADL, since these evaluate complex activities and social interactions, with the possibility of determining changes over time [18,26], although this has not been well explored in community-dwelling older adults. This study showed how community-dwelling older adults with appropriate cognitive functioning perform complex and social activities, although we found that it depended on the age, sex, and other sociodemographic factors, as well as lifestyle and health condition variables. Furthermore, this research established the risk factors that limit the performance of ≥3 AADLs, and is, to our knowledge, the first study with this approach; the research about AADLs has been carried out on older adults with cognitive impairments or depression [14,15,19,21,29].

In this sense, the HA concept is often used to describe a positive disease-free state, and distinguishes between healthy and sick individuals; however, this should not be applied in older people, as many of them may have one or more well-controlled chronic NCDs, but live independent and autonomous lives, with well-being and quality of life. Therefore, the WHO in the 2015 World Report on Aging and Health proposed that an older adult with HA must have a functional capacity that allows well-being in old age. This functional level must be determined, not only with an evaluation of physical and cognitive capacities, but also with the interactions these older adults have with their environment [13]. Thus, the WHO proposes integrated care for older people, namely the ICOPE strategy, specifying the guidelines for identifying priority conditions associated with the reduction in intrinsic capacity and the environment, to maintain, extend, or restore functional capacity, and promote HA in the community, with a focus on person-centered care, and not on isolated illnesses or symptoms [30]. Moreover, it is necessary first to determine the status of functional capacity of older people in the community. This was the aim of this study.

In the MHAS 2018 wave, the ratio of older adults without cognitive impairment was 70%, slightly higher than the results reported by a previous study with the same database but using the 2015 wave data, which found that more than 35% of the adults aged 60 and over had cognitive impairment [8]. Another study with Mexican elderlies found that 83% retained their cognitive function [31], corroborating that community-dwelling older Mexican adults maintain that functionality. Similarly, a study that used harmonized datasets from national surveys of Japan, Korea, and China found the highest ratio in Japan (90.8%) and Korea (75.1%), but not in China (44.2%) [32].

In this study, some MHAS questions were selected to measure AADLs and, although KR-20 values over 0.6 indicate that items in a test are homogeneous [33], the consistency values of the proposed AADL constructs here ranged between 0.4 and 0.6; it can be considered that these questions are consistent with what they intend to measure, because the survey used is not a test per se. Indeed, a list of 13 AADLs in a questionnaire format was validated and showed high internal consistency, with an alpha coefficient of 0.80 [26]. Nine of these activities were used in this research, maintaining the same domains; hence, the MHAS questions selected are representative of AADLs. Moreover, the research that used national surveys of East Asia selected community, social clubs, and leisure activities, besides volunteering and paid work, as active engagement [32], items like in our study. 

In this regard, 2363 (37%) of the older adults without cognitive impairments performed ≥ 3 AADLs, mostly men (40%), contrary to a study with Peruvian older adults without neurocognitive disorders that reported an independence of 80% to carry out AADLs [29], and people of 65–75 years from China (73.5%) and Korea (55.9%) [32]. This difference may be due to the criteria used for defining dependence for AADLs and age.

From all the evaluated activities, the social domain was performed by the older adults most frequently, mainly attending religious services (76%) and talking with relatives and friends (75%). These results are in accordance with the previous report of the 2015 MHAS that showed social engagement in more than 80% of people aged 60 and over [8]; this was higher than the Health, Wellbeing, and Aging Study (SABE) in Jalisco, Mexico, with 48% of older adults participating in social activities [31]. Therefore, it is well known that social activity has a positive impact on global cognition; this suggests that these activities may promote brain health in older adults [34] and better performance on tests of executive functions [19]. On the opposite site, social vulnerability is related with mortality and incident disability, which was reported in a longitudinal study from the 2012 and 2015 MHAS data [20]. It is important to highlight that women mainly have social activities, while men have productive activities. This is due to their gender roles in society that are associated with cultural practices; for older Mexican adults, the man is the provider, and the woman is the one who performs caregiving activities [21]. Moreover, it is noteworthy that older adults with healthy cognition do not distinguish between doing physical or leisure activities, probably because they focus on other activities.

On the other hand, Mexican people are very religious; hence, 74% of people from the MHAS 2015 wave think that the religion is very important, hence the high rate of participation in religious events [35]; this is contrary to older Indian adults who are very religious (79%), but do not participate in religious activities [36]. Notwithstanding this difference, religion as a social activity has been associated with healthier biological functioning and a lower risk of all-cause mortality [35,37]; in addition, it reduces the chance of cognitive impairment, mainly in older adults with depression [36]. Religious events are an important social activity in this stage of life. 

Regarding sociodemographic variables, as age increases, the total AADLs performed by the participants decreases. On the contrary is the educational level, in which less educated older people carry out fewer AADLs, as is evident in illiterate older adults. Likewise, the risk of performing fewer than three AADLs increases with age; elders aged 90 years and older are almost eight times more at risk of carrying out fewer AADLs, and 82% of people aged 60 years and older without schooling perform fewer AADLs, independent of sex. It is probable that the diminished total of AADLs performed by aged adults is related to the functional incapacities that appear through time, such as physical or cognitive decline, mobility problems, impairments from NCDs, or depressive symptoms [8,20,27]. Recent studies have confirmed that changes in gait speed and body balance are negatively associated with falls; moreover, there are age-related sensorimotor deficits that show reduced dexterity among older adults; although they develop strategies to adapt, compensate, and correct deficits [16,17], these physiological changes probably limit the possibilities of performing physical/leisure activities, which was observed in this research.

Regarding the educational level, there are no reports of the relationship between this and AADLs; however, two studies with older Mexican adults have shown that a high education level is weakly related to HA, based on Rowe and Kahn’s model [21,22], probably because older people with more education try to stay active through knowing the health benefits of it.

Another related variable is gender, which shows that 18% of women have the most trouble performing ≥ 3 AADLs. This is consistent with previously reported studies in the old Mexican population with HA [8,21,22], and may be because the women have lower physical function and less productive participation, also shown in this study.

Urban residence is a risk factor for performing ≥ 3 AADLs; this is new knowledge, because there are no previous reports about it. A recent investigation about successful aging (SA) pointed out an inconsistency in the comparison between rural and urban areas of China and Korea. In China, the proportion of individuals with SA is lower in rural areas, contrary to Korea, which presented a higher prevalence for those living in urban areas [32]. In Mexico, rural populations often carry out more activities, especially productive activities with high physical demands, in order to meet their economic needs, which is not common in urban areas; therefore, as mentioned above, only in men is urban residence a risk factor.

In relation to lifestyle factors, it is well recognized in both cross-sectional and longitudinal studies that physical activity is a beneficial factor for health, with the possibility of performing more AADLs, and is especially associated with HA; this was corroborated in this study [19,22,38,39,40]. Alcohol consumption is a controversial factor because several studies show that it is associated with SA, HA, and other health benefits like this research [22,38,39,40]. Indeed, a systematic review with meta-analysis showed that there is a beneficial association of reasonable alcohol intake with HA compared with non-drinkers [41]; however, this is not comparable with our results because the data were self-reported, and it was not possible to determinate a “reasonable intake”, since it was only a yes/no question; thus, information bias is possible. This is why this result should be interpreted with caution.

The last factor related to performing fewer than three AADLs is having more than two chronic NCDs. Although the possibility of not being able to perform ≥ 3 AADLs depends on the type of disease that the older adult is suffering from, previous research agrees that with an increase in the number of NCDs, there is less chance of having more activity [8,19,29]. This is noticeable in men, probably due to cultural practices, since Mexican women tend to adapt better to diseases.

This study has some limitations. Firstly, this is a secondary analysis of a database; therefore, the data were obtained and captured by people outside the investigation group, and were beyond our control. Furthermore, the survey was self-reported; thus, it was not possible to confirm several answers, such as diseases of the elderly, because sometimes older adults are unaware of their health conditions. Another limitation is the criteria used to define the AADLs; as previously stated, data were obtained from selected questions of the survey, and they did not cover all the proposed AADLs, although the same domains were used to maintain the essence of the activities. Despite these limitations, a strength was that the database contains national information, with representation from all sectors of Mexican society. In addition to the large sample size that would otherwise be difficult to collect, the results have a national scope. This study is the first to show not only the frequency with which older adults without cognitive damage carry out complex activities, but the factors associated with the failure to execute them.

## 5. Conclusions

Our findings suggest a low prevalence of older adults being able to perform three or more AADLs, with a difference by sex. Likewise, factors associated with not performing more than three activities are age, gender, education level, urban residency, sedentary lifestyle, and suffering from two or more chronic NCDs; however, longitudinal studies are necessary to confirm our results. Furthermore, it is necessary to encourage the active participation of older people, as well as their families and community in the implementation of the ICOPE strategy [24] at the community level, focusing on functional older people, to maintain and promote the execution of AADLs, and thereby promote healthy aging.

## Figures and Tables

**Figure 1 healthcare-11-02107-f001:**
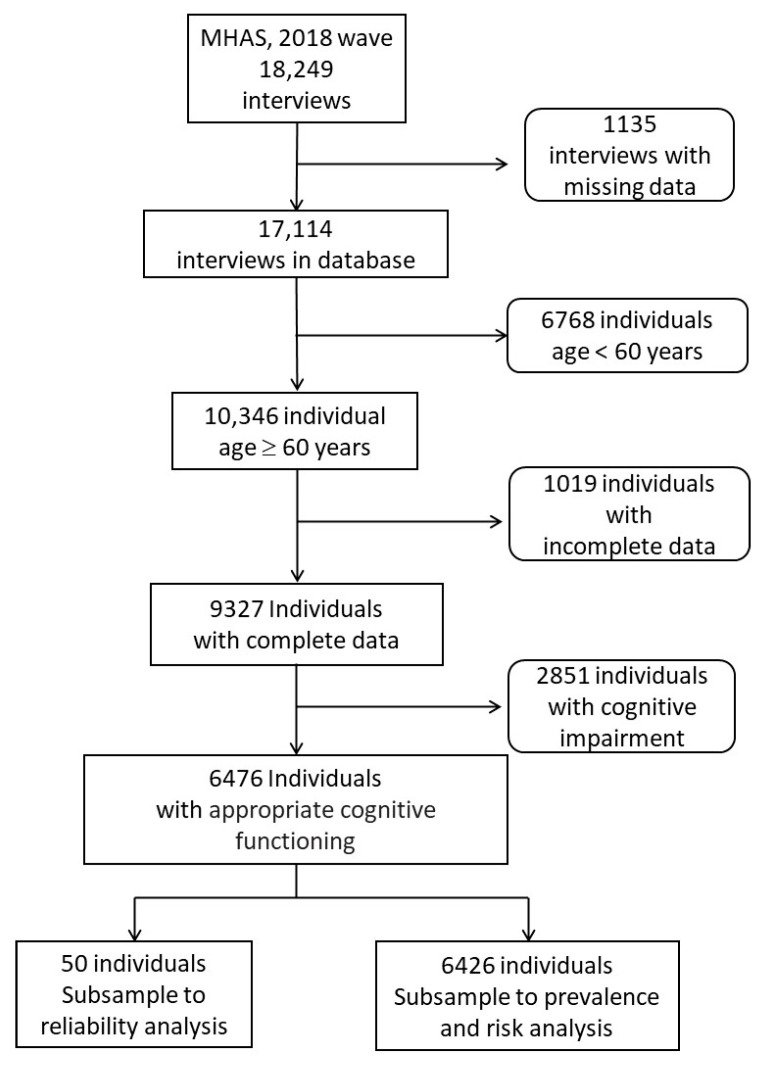
Flow chart of sample selection.

**Table 1 healthcare-11-02107-t001:** Advanced activities of daily living of the available questions from the Mexican Health and Aging Study, 2018 wave. These were chosen from previous proposals [15,26,27].

Physical/Leisure Domain	Social Domain	Productive Domain
Assist with a sport or social club	Do you attend religious services?	In the last 2 years, did you participate in any volunteer work for a religious, educational, charity organization, or for the community?
Assist in a lecture, seminar, or class	Talk to relatives or friends on the phone or use the computer to send emails or use the Internet	Work as a volunteer or help with a non-profit organization without pay or compensation.
Sew, embroider, knit, or other crafts	Did you take care of a sick family member between 2015 and 2018?	During the last year, did you have a primary paid job?

**Table 2 healthcare-11-02107-t002:** Sociodemographic characteristics, health conditions, and lifestyle habits of the total population analyzed and stratified by sex; from 2018 MHAS data.

Characteristic	Total (n = 9327)	Men (n = 4108, 44%)	Women (n = 5219, 56%) *
Age (years)	71.5 ± 7.8	71.8 ± 7.8	71.1 ± 7.8
Age group			
60–69 years	4135 (44%, 43–45%)	1682 (41%, 39–42%)	2453 (47%, 46–48%) *
70–79 years	3659 (39%, 38–40%)	1729 (42%, 41–44%)	1930 (37%, 36–38%) ^†^
80–89 years	1359 (15%, 14–16%)	613 (15%, 14–16%)	746 (14%, 13–15%)
90–99 years	168 (1.9%, 1.5–2.1%)	80 (2%, 1.5–2.4%)	88 (1.7%, 1.3–2%)
≥100 years	6 (0.1%, 0–0.12%)	4 (0.1%)	2 (< 0.1%)
Years of education	5.5 ± 4.7	6.1 ± 5.0	5.0 ± 4.3 ^a^
Educational level ^b^			
Illiterate	1630 (18%, 17–19%)	625 (16%, 14–17%)	1005 (19%, 18–20%) ^‡^
Elementary	4954 (53%, 52–55%)	2133 (53%, 51–54%)	2821 (54%, 53–56%)
Middle school	1461 (16%, 15–17%)	569 (14%, 13–15%)	892 (17%, 16–18%) ^‡^
High school	368 (4%, 3.6–4.4%)	211 (5%, 4–6%)	157 (3%, 2.6–3.5%)
College or higher	822 (9%, 8–10%)	504 (12%, 11–13%)	318 (6%, 5–7%) *
Locality size			
Population ≥ 100,000 inhab.	5301 (57%, 56–58%)	2249 (55%, 53–56%)	3052 (59%, 57–60%)
Population 15,000 y 99,999 inhab.	1269 (13%, 12–14%)	540 (13%, 12–14%)	729 (14%, 13–15%)
Population 2500 y 14,999 inhab.	895 (10%, 9–11%)	405 (10%, 9–11%)	490 (9%, 8.6–10%)
Population < 2500 inhab.	1862 (20%, 19–21%)	914 (22%, 21–24%)	948 (18%, 17–19%)
Marital status			
Married/in a civil union	5883 (63%, 62–64%)	3248 (79%, 78–80%)	2635 (50%, 49–52%) *
Never married	430 (5%, 4–6%)	122 (3%, 2.5–3.5%)	308 (6%, 5–7%) ^‡^
Divorced/separated	669 (7%, 6–8%)	213 (5%, 4–6%)	456 (9%, 8–10%) ^§^
Widowed	2345 (25%, 24–26%)	525 (13%, 12–14%)	1820 (35%, 34–36%) *
Health condition			
With NCD	6323 (68%, 67–69%)	2448 (60%, 58–61%)	3912 (75%, 74–76%) *
Healthy	2964 (32%, 31–33%)	1657 (40%, 39–42%)	1307 (25%, 24–26%) *
One NCD	3629 (39%, 38–40%)	1513 (37%, 35–38%)	2116 (41%, 39–42%) ^‡^
Two NCD	2243 (24%, 23–25%)	780 (19%, 18–20%)	1468 (28%, 27–29%) *
≥3 NCD	491 (5%, 4–6%)	164 (4%, 3–5%)	328 (6%, 5.6–7%)
Mild cognitive impairment	2425 (26%, 25–27%)	575 (14%, 13–15%)	1983 (38%, 37–39%) *
Severe cognitive impairment	373 (4%, 3.6–4.4%)	41 (1%, 0.7–1.3%)	313 (6%, 5–7%) *
Lifestyle			
Smoking	919 (10%, 9–11%)	648 (16%, 15–17%)	271 (5%, 4.6–6%) *
Alcohol intake	2247 (24%, 23–25%)	1496 (36%, 35–38%)	751 (14%, 13–15%) *
Physical activity	2717 (29%, 28–30%)	1548 (38% 36–39%)	1169 (22%, 21–24%) *
Average of performed AADLs	5.67 ± 1.30	5.60 ± 1.70	5.75 ± 1.27 ^a^

Chi-square test * *p* < 0.0001; ^†^
*p* = 0.001; ^‡^
*p* < 0.05; ^§^
*p* < 0.01; ^a^ independent *t*-test, *p* < 0.0001; ^b^ missing data (n = 92). NDC: chronic noncommunicable disease. Quantitative data are means ± SD, categorical data are frequencies (%, 95% CI).

**Table 3 healthcare-11-02107-t003:** Kuder-Richardson test for the advanced activities of daily living included in this study, and sensibility analysis. From 2018 MHAS data.

Component	KR-20
Total	0.594 *
Removing	
Attend religious services	0.575 *
Talk to relatives or friends	0.546 *
Sew, embroider, knit, or other crafts	0.589 *
Care of a sick family member	0.587 *
Assist with a sport or social club	0.558 *
Assist in a lecture, seminar, or class	0.525 *
Participate in any volunteer work for the community	0.466 *
Work as a volunteer without pay or compensation	0.554 *
Have a primary paid job	0.611 *

KR-20: Kuder-Richardson 20 test. * *p* < 0.0001.

**Table 4 healthcare-11-02107-t004:** Prevalence of each advanced activity of daily living, and mean of the activities performed by domain in total older adults without cognitive impairment, stratified by sex. From 2018 MHAS data.

Activity	Total (n = 6476)	Men (n = 3491, 54%)	Women (n = 2985, 46%)
Physical/leisure domain	0.04 ± 0.21	0.06 ± 0.23	0.03 ± 0.17
Assist with a sport or social club (n = 6474)	288 (4%, 3–5%)	202 (6%, 5–7%)	86 (3%, 2–4%) *
Assist in a lecture, seminar, or class (n = 6473)	0	0	0
Sew, embroider, knit, or other crafts (n = 6453)	0	0	0
Social domain	1.69 ± 0.76	1.60 ± 0.77	1.78 ± 0.73 ^†^
Do you attend religious services? (n = 6471)	4922 (76%, 75–77%)	2515 (72%, 71–74%)	2407 (81%, 79–82%) *
Talk to relatives or friends on the phone, etc. (n = 6473)	4825 (75%, 73–76%)	2518 (72%, 71–74%)	2307 (77%, 76–79%) *
Did you take care of a sick family member (n = 6327)	1166 (18%, 17–19%)	553 (16%, 15–17%)	613 (21%, 19–22%) *
Productive domain	0.52 ± 0.73	0.67 ± 0.78	0.36 ± 0.63 ^†^
Did you participate in any volunteer work in community? (n = 6471)	929 (14%, 13–15%)	497 (14%, 13–15%)	432 (15%, 13–16%)
Work as a volunteer without pay (n = 6475)	459 (7%, 6–8%)	288 (8%, 7–9%)	171 (6%, 5–6.6%) *
Did you have a primary paid job? (n = 6452)	2005 (31%, 30–32%)	1538 (44%, 43–46%)	467 (16%, 14–17%) *
Mean of AADLs performed	2.25 ± 1.15	2.32 ± 1.21	2.17 ± 1.07 ^†^
Frequency of AADLs performed			
≥3 AADLs	2363 (37%, 35–38%)	1410 (40%, 39–42%)	953 (32%, 30–34%) *
<3 AADLs	4113 (63%, 62–65%)	2081 (60%, 58–61%)	2032 (68%, 66–70%)

* Chi-square test, *p* < 0.0001; ^†^ independent *t*-test, *p* < 0.0001. AADLs: advanced activities of daily living. Note: the sample size is different in each item due to lack of information. Quantitative data are means ± SD, categorical data are frequencies (%, 95% CI).

**Table 5 healthcare-11-02107-t005:** Numbers of advanced activities of daily living according to sociodemographic, lifestyle, and health characteristics of the total sample, stratified by sex. Data show mean ± standard deviation. From 2018 MHAS data.

Characteristic	Total (n = 6476)	Men (n = 3491)	Women (n = 2985)
Age group			
60–69 years (n = 2675)	2.55 ± 1.18	2.67 ± 1.22	2.43 ± 1.12 ^‡^
70–79 years (n = 2594)	2.18 ± 1.11 ^a^	2.22 ± 1.18	2.12 ± 1.01 ^§^
80–89 years (n = 1056)	1.78 ± 0.97 ^a,c^	1.83 ± 1.03	2.25 ± 1.27 ^‡^
90–99 years (n = 145)	1.49 ± 0.88 ^a,c,d^	1.62 ± 0.95	1.73 ± 0.88
≥100 years (n = 6)	1.00 ± 0.89 ^b^	0.75 ± 0.96	1.50 ± 0.71
Locality size			
Rural < 2500 inhab. (n = 1345)	2.38 ± 1.24	2.57 ± 1.27	2.10 ± 1.14 ^‡^
Urban ≥ 2500 inhab. (n = 5131)	2.22 ± 1.13 *	2.25 ± 1.18	2.19 ± 1.06
Educational level (n = 9183)			
Illiterate (n = 1251)	2.00 ± 1.14	2.08 ± 1.21	1.92 ± 1.08 ^§^
Elementary (n = 3502)	2.26 ± 1.13 ^e,g,h^	2.34 ± 1.21	2.17 ± 1.02 ^‡^
Middle school (n = 935)	2.41 ± 1.19 ^e^	2.42 ± 1.25	2.40 ± 1.13
High school (n = 231)	2.32 ± 1.00 ^f^	2.32 ± 1.02	2.34 ± 0.97
College or higher (n = 481)	2.55 ± 1.21 ^e^	2.54 ± 1.21	2.56 ± 1.22
Marital status			
Married/in a civil union (n = 4230)	2.33 ± 1.17	2.45 ± 1.42	2.25 ± 1.09 ^‡^
Never married (n = 281)	2.18 ± 1.11	2.62 ± 1.38	2.19 ± 1.09 ^¶^
Divorced/separated (n = 424)	2.23 ± 1.12	2.54 ± 1.34	2.20 ± 1.07 ^¶^
Widowed (n = 1541)	2.07 ± 1.09 ^i^	2.27 ± 1.29	2.06 ± 1.05 ^¶^
Smoking			
Yes (n = 712)	2.36 ± 1.24 ^†^	2.39 ± 1.28	2.26 ± 1.07
No (n = 5761)	2.24 ± 1.14	2.31 ± 1.20	2.17 ± 1.07 ^‡^
Alcohol intake			
Yes (n = 1652)	2.49 ± 1.22 *	2.51 ± 1.25	2.44 ± 1.11
No (n = 4823)	2.17 ± 1.12	2.22 ± 1.18	2.13 ± 1.06 ^¶^
Physical activity			
Yes (n = 1837)	2.69 ± 1.21 *	2.73 ± 1.23	2.60 ± 1.18 ^§^
No (n = 4633)	2.08 ± 1.08	2.09 ± 1.14	2.07 ± 1.03
Health condition			
Healthy (n = 2113)	2.36 ± 1.21	2.45 ± 1.23	2.19 ± 1.14 ^‡^
1 NCD (n = 2518)	2.27 ± 1.16 ^j^	2.35 ± 1.25	2.19 ± 1.06 ^‡^
2 NCDs (n = 1510)	2.09 ± 1.06 ^k,m^	2.03 ± 1.06	2.14 ± 1.06 ^§^
≥3 NCDs (n = 333)	2.15 ± 1.05 ^l^	2.14 ± 1.07	2.16 ± 1.03

Independent *t*-test, * *p* < 0.0001; ^†^
*p <* 0.05. ^a^ One-way ANOVA with Tukey’s post hoc test, 60–69 vs. 70–79, vs. 80–89, and vs. 90–99, *p <* 0.0001; ^b^ 60–69 vs. ≥ 100, *p <* 0.001; ^c^ 70–79 vs. 80–89 and vs. 90–99, *p <* 0.0001; ^d^ 80–89 vs. 90–99, *p <* 0.05; ^e^ illiterate vs. elementary, vs. middle school, and vs. college, *p <* 0.0001; ^f^ illiterate vs. high school, *p =* 0.001; ^g^ elementary vs. middle school, *p <* 0.001; ^h^ elementary vs. college, *p* < 0.0001; ^i^ married vs. widowed, *p* < 0.0001; ^j^ healthy vs. 1 NCD, *p* < 0.05; ^k^ healthy vs. 2 NCDs, *p* < 0.0001; ^l^ healthy vs. ≥ 3 NCDs; ^m^ 1 vs. 2 NCDs. Independent *t*-test, men vs. women, ^‡^
*p* < 0.0001; ^§^
*p* < 0.05, ^¶^
*p* < 0.01. NCD: chronic noncommunicable disease. Note: the sample size is different in each item due to lack of information.

**Table 6 healthcare-11-02107-t006:** Factors associated with <3 advanced activities of daily living in a multivariate analysis, total and stratified by sex. From 2018 MHAS data.

Factor		Total			Women			Men	
	OR *	95% CI	*p*-Value	OR ^†^	95% CI	*p*-Value	OR ^‡^	95% CI	*p*-Value
Age group									
60–69 years ^a^	1								
70–79 years	1.66	1.47–1.86	<0.0001	1.58	1.32–1.89	<0.0001	1.75	1.49–2.06	<0.0001
80–89 years	3.62	2.99–4.38	<0.0001	3.74	2.80–5.00	<0.0001	3.71	2.88–4.77	<0.0001
≥90 years	8.75	4.77–16.03	<0.0001	20.23	4.89–83.68	<0.0001	6.68	3.34–13.36	<0.0001
Sex (women)	1.18	1.04–1.34	0.0008						
Locality size									
Rural < 2500 inhab. ^a^	1								
Urban ≥ 2500 inhab.	1.36	1.18–1.56	<0.0001	0.90	0.72–1.12	0.337	1.79	1.49–2.15	<0.0001
Educational level									
College or higher ^a^	1								
Illiterate	1.82	1.43–2.81	<0.0001	1.87	1.23–2.84	0.003	1.89	1.39–2.56	<0.0001
Elementary	1.41	1.15–1.73	0.001	1.50	1.02–2.20	0.039	1.44	1.12–1.84	0.004
Middle school	1.22	0.96–1.54	0.098	1.26	0.84–1.91	0.266	1.25	0.93–1.68	0.137
High school	1.74	1.24–2.45	0.001	1.81	0.96–3.42	0.069	1.71	1.15–2.56	0.008
Marital status									
Married/in a civil union ^a^	1								
Never married	1.02	0.78–1.33	0.894	0.83	0.59–1.18	0.307	1.48	0.96–2.28	0.074
Divorced/separated	1.10	0.88–1.37	0.409	1.10	0.82–1.49	0.516	1.09	0.78–1.54	0.599
Widowed	1.74	1.24–2.45	0.001	0.90	0.75–1.09	0.299	0.86	0.69–1.09	0.221
Sex (women)	1.18	1.04–1.34	0.0008						
Sedentary	2.15	1.91–2.42	<0.0001	2.17	1.79–2.64	<0.0001	2.09	1.79–2.43	<0.0001
Smoking	1.04	0.88–1.24	0.659	1.01	0.72–1.43	0.953	1.08	0.88–1.32	0.446
Alcohol intake	0.79	0.70–0.90	<0.0001	0.71	0.57–0.90	0.004	0.85	0.72–0.99	0.032
Health condition									
Healthy/one disease ^a^	1								
≥2 diseases	1.35	1.19–1.53	<0.0001	1.08	0.91–1.28	0.388	1.71	1.43–2.06	<0.0001

Logistic regression * R^2^ = 14.8%, *p* < 0.0001; ^†^ R^2^ = 12.3%; ^‡^ R^2^ = 16.9%, *p* < 0.0001. ^a^ Control group.

## Data Availability

The data presented in this study are available on request from the corresponding author. Original data files and documentation are for public use and are available at www.MHASweb.org (accessed on 2 October 2020).

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
