# Peer review of "Advanced Activities of Daily Living in Community-Dwelling Older Adults: A Cross-Sectional Study of the Mexican Health and Aging Study (MHAS 2018)"

_healthcare, 2023, doi:10.3390/healthcare11142107_

Round 1

Reviewer 1 Report

Congratulations to the authors, given the novelty of the study, but allow me to point out a minor modification

Introduction, line 41: Frailty and disability are common terms in the vocabulary of professionals involved in the care of the elderly population. In the medical literature, the most commonly used criterion for detecting or classifying an elderly person as frail has been the presence of difficulty in performing activities of daily living. Although there seems to be a certain consensus that frailty is a clinical entity that encompasses other symptoms in addition to disability, there is also a notable clinical perception that disability is not an irremediable consequence of frailty, so it would be necessary to clarify both concepts in the introduction". Within the heterogeneity of ageing, not all older adults are frail" 

Author Response

Introduction, line 41: Frailty and disability are common terms in the vocabulary of professionals involved in the care of the elderly population. In the medical literature, the most commonly used criterion for detecting or classifying an elderly person as frail has been the presence of difficulty in performing activities of daily living. Although there seems to be a certain consensus that frailty is a clinical entity that encompasses other symptoms in addition to disability, there is also a notable clinical perception that disability is not an irremediable consequence of frailty, so it would be necessary to clarify both concepts in the introduction". Within the heterogeneity of ageing, not all older adults are frail" 

Thank you for reviewing our research and your suggestion is valuable for our paper. Attached please find the revised version, your observation has been acknowledged and the modification is highlighted in blue, the concept has been clarified, and one reference has been added (lines 41-46).

Reviewer 2 Report

The manuscript entitled “Advanced Activities of Daily Living in Community-Dwelling Older Adults: A Cross-Sectional Study of the Mexican Health and Aging Study (MHAS 2018)" by Sánchez-Rodríguez et al is informative study and presented finely. The study has addressed the response of older population towards advanced activity of daily living in community.  However, following concerns need to be addressed and reconciled which could improve/upgrade this manuscript.

·       Abstract is not properly described by the authors. Author should rewrite the abstract, must briefly write about background, objective, method and their outcomes clearly. 

·       Smoking and drinking alcohol declined the cognitive ability above the recommended limits. So, what is the selection criteria for limit of smoking and alcohol intake?

·       In the selected study years, why there is too long gap in early years without any continuity. Any specific reason to choose different years for interviews?

·       Did authors find out any difference in pre and post covid (pandemic) period in the cognitive status of older population.

·       In the manuscript authors study was according to the 2018 wave of the Mexican Health and Aging Study (MHAS) and the findings were from 2001. Authors musts explain the connectivity.

·       I suggest to the authors to show his notably important finings data in graphical format to enhance the visibility of their manuscript.

Thanks

Good writing quality and English language

Author Response

Thank you for reviewing our research and your suggestion is valuable for our paper. We have checked our manuscript carefully and the revisions are listed below. Attached please find the revised version, your observations have been acknowledged and the modifications are highlighted in green.

1. Abstract is not properly described by the authors. Author should rewrite the abstract, must briefly write about background, objective, method and their outcomes clearly.

The abstract was revised and changed (lines 13-26).  

2. Smoking and drinking alcohol declined the cognitive ability above the recommended limits. So, what is the selection criteria for limit of smoking and alcohol intake?

Thank you very much for your comment. In our study, we only included older adults without cognitive impairment. Smoking and alcohol intake were variables included as possible confounders and for descriptive purposes. Only 712/6476 older adults were smokers and 1652/6476 drink alcohol. The survey data was based on whether the interviewers smoke or drink alcohol currently, not on the number of cigarettes or drinks they consume.

3. In the selected study years, why there is too long gap in early years without any continuity. Any specific reason to choose different years for interviews?

Thank you very much for your comment. We think that there is a confusion since we have only included the data from the 2018 wave. For the reason, we have eliminated lines to avoid confusion (lines 111-114)

4. Did authors find out any difference in pre and post covid (pandemic) period in the cognitive status of older population.

It is a cross-sectional study from MHAS 2018 wave. The surveys were conducted before the COVID-19 pandemic, there is no possibility to do this comparison.

5. In the manuscript authors study was according to the 2018 wave of the Mexican Health and Aging Study (MHAS) and the findings were from 2001. Authors musts explain the connectivity

We apologize for any confusion, but we did not analyze 2001 data. In the methods section and all the tables of results, we specify that the data are from the 2018 wave.

6. I suggest to the authors to show his notably important finings data in graphical format to enhance the visibility of their manuscript.

We appreciate the suggestion; we will take it in account.

Reviewer 3 Report

The article by Sanchez-Rodriguez et al focusses on the advanced activities of daily living (AADL) in community-dwelling older adults, specifically from the MHAS 2018 sample. The authors analyzed the MHAS 2018 data and identified factors associated with execution of AADL, assessed its disparity between two sexes, and point to potential differences in lifestyle, education level, locality, and other factors that could possibly explain the observed trends.

Overall, the study is aimed at addressing an important question on effects of aging on personal well-being. The authors employ a systematic approach to address the same. My primary concerns are mostly surrounding the novelty of this study, missing details in statistical analysis, and insufficient description of physiological processes that could underlie deficits in AADL with aging – all of which, I believe, could be addressed by incorporating detailed description and clarification at specified instances. I have highlighted my concerns below:

1.     It is unclear how the current study is different from those published using the MHAS 2015 sample. It might help the readers contextualize novelty of this article by clearly stating the research gaps not focused by the previous studies analyzing MHAS 2015 and 2018 data but are addressed by the current study.

2.     Are the older adults (who were assessed for AADL) already assessed to have higher performance in BADL and IADL scores? Is it possible that older adults who could perform > 3 AADL may not be able to perform > 3 BADL and/or IADL? The interdependency between these three types of ADLs need to be described in the introduction section.

3.     Lines 67-71: While the authors acknowledge previous studies focusing on cognitive decline following aging, it would help provide readers a physiological basis of aging affecting complex sensorimotor functions. For instance, following recent studies showed more nuanced sensorimotor deficits in tasks involving upper and lower extremities, pointing to potential mechanisms due to which one would expect impairment in AADL with aging:

Article 1: Nascimento MdM, Gouveia ÉR, Gouveia BR, Marques A, Martins F, Przednowek K, França C, Peralta M, Ihle A. Associations of Gait Speed, Cadence, Gait Stability Ratio, and Body Balance with Falls in Older Adults. International Journal of Environmental Research and Public Health. 2022; 19(21):13926. https://doi.org/10.3390/ijerph192113926  

Article 2: Rao N, Mehta N, Patel P, Parikh PJ. Effects of aging on conditional visuomotor learning for grasping and lifting eccentrically weighted objects. J Appl Physiol (1985). 2021 Sep 1;131(3):937-948. doi: 10.1152/japplphysiol.00932.2020

Findings from these studies could be cited to provide essential physiological perspective to why older adults may exhibit lower scores in their physical/leisure domain activities involving standing and balance control tasks (e.g., assisting with sports or social club/lecture/class activities), and those involving skillful control of finger movements (e.g., sewing, embroidering, knitting, etc.). This would help authors’ rationale for expecting deficits in AADL with aging.

4.     Lines 134-138: Were any covariates used in the multiple logistic regression analysis? For instance, lifestyle of a person could be a function of age, education level and marital status. Similarly, health condition could be a function of age, education level and lifestyle, just to name a few. Did the authors assess such associations in the current sample, and attempt addressing such interdependencies? If so, the findings can be reported, and if not, this needs to be highlighted in the limitations section.

5.     In table 3, it is counterintuitive to expect a significant increment in KR-20 score following the removal of ‘Having a primary paid job’ question. A description to potential reasons for this finding is needed to guide future studies.

6.     Lines 176-177: It is unclear as to which statistical test was used to compare the number of men performing ≥ 3 AADL with the number of women in the same condition, especially when this metric would result in a fixed number without any error bars. A clarification is essential.

7.     Lines 189-191: Statistical comparison of the findings here seems to be missing. Please rectify the same.

8.     Table 6 has several of its rows truncated/unreadable perhaps due to its size within a page. The authors may consider splitting the presentation of table content into two pages.

9.     In lines with my comment for the introduction section, the discussion section could benefit from relating physiological effects of aging on upper and lower limb functions with difficulties observed among older adults in performing AADLs.

10.  A careful proofreading of the manuscript is needed to address occasional typos or grammatical errors, for e.g.,

line 48: ‘… aspect to considerate…’

line 68: ‘… which are loss initially…’

line 96: ‘… do not were directly include.’

line 111: ‘… as times as questions were included…’

line 382: ‘… with do not perform these …’

As mentioned in my comments, correction of occasional typos/grammatical errors and careful proofreading is needed in the manuscript.

Author Response

Thank you for reviewing our research and your suggestion is valuable for our paper. We have checked our manuscript carefully and the revisions are listed below. Attached please find the revised version, your observations have been acknowledged and the modifications are highlighted in yellow.

1. It is unclear how the current study is different from those published using the MHAS 2015 sample. It might help the readers contextualize novelty of this article by clearly stating the research gaps not focused by the previous studies analyzing MHAS 2015 and 2018 data but are addressed by the current study.

We thanks to the reviewer for this observation. A paraph with additional information was included (lines 84-102).

2. Are the older adults (who were assessed for AADL) already assessed to have higher performance in BADL and IADL scores? Is it possible that older adults who could perform > 3 AADL may not be able to perform > 3 BADL and/or IADL? The interdependency between these three types of ADLs need to be described in the introduction section.

We thank for this comment. A paraph with additional information was included (lines 68-72)

3. Lines 67-71: While the authors acknowledge previous studies focusing on cognitive decline following aging, it would help provide readers a physiological basis of aging affecting complex sensorimotor functions. For instance, following recent studies showed more nuanced sensorimotor deficits in tasks involving upper and lower extremities, pointing to potential mechanisms due to which one would expect impairment in AADL with aging. Findings from these studies could be cited to provide essential physiological perspective to why older adults may exhibit lower scores in their physical/leisure domain activities involving standing and balance control tasks (e.g., assisting with sports or social club/lecture/class activities), and those involving skillful control of finger movements (e.g., sewing, embroidering, knitting, etc.). This would help authors’ rationale for expecting deficits in AADL with aging.

Your help is valued. We have examined the papers and added information to support the potential physiological role of aging in performing AADL (lines 74-77)

4. Lines 134-138: Were any covariates used in the multiple logistic regression analysis? For instance, lifestyle of a person could be a function of age, education level and marital status. Similarly, health condition could be a function of age, education level and lifestyle, just to name a few. Did the authors assess such associations in the current sample, and attempt addressing such interdependencies? If so, the findings can be reported, and if not, this needs to be highlighted in the limitations section.

We appreciate your observations because it seems that the paraph is unclear. We include all the sociodemographic, lifestyle and health conditions factors in the multivariate logistic models. To clarify this fact, the paraph was modified (lines 162-167). The results presented in table 6 are from these models with all the variables.  

5. In table 3, it is counterintuitive to expect a significant increment in KR-20 score following the removal of ‘Having a primary paid job’ question. A description to potential reasons for this finding is needed to guide future studies. 

We thank you comment. This apparent contradiction is probably due to the difference between doing or not doing the activity, since in the case of “having a paid job” it is very small, compared to the other activities (difference between 60 and 88%). This affects the calculation of KR20, showing an apparent inconsistency of this activity as part of the construct.  A possible explanation was added (lines 197-199)

6. Lines 176-177: It is unclear as to which statistical test was used to compare the number of men performing ≥ 3 AADL with the number of women in the same condition, especially when this metric would result in a fixed number without any error bars. A clarification is essential.

Your comment is appreciated. The statistical test was incorrectly indicated. Chi squared was used as a comparison test because the results are frequencies. The asterisk in the table was misplaced because it is in the p value and not in the statistical test. The asterisk was moved to where it belongs (line 217). We agree that the measure of uncertainty of the proportions is necessary, thus, we add the 95% confidence interval for the better appreciation of the results in tables 2 and 4.

7. Lines 189-191: Statistical comparison of the findings here seems to be missing. Please rectify the same.

We thank for your comment, it was an unforgivable omission. The information has been added (line 224)

8. Table 6 has several of its rows truncated/unreadable perhaps due to its size within a page. The authors may consider splitting the presentation of table content into two pages.

We appreciate your concern; this is a matter of editing. In the submitted manuscript, the table occupies a single page, and it is not truncated, perhaps it was modified when it sent for review. Reviewing the document sent for review, we noticed this detail, so the table was divided into two pages.

9. In lines with my comment for the introduction section, the discussion section could benefit from relating physiological effects of aging on upper and lower limb functions with difficulties observed among older adults in performing AADLs.

We are grateful for your recommendation, the information from both papers was incorporate to improve the manuscript (lines 381-386)

10. A careful proofreading of the manuscript is needed to address occasional typos or grammatical errors, for e.g.,

line 48: ‘… aspect to considerate…’ The text was changed (lines 53-54)

line 68: ‘… which are loss initially…’ The text was changed (lines 80-81)

line 96: ‘… do not were directly include.’ The text was removed.

line 111: ‘… as times as questions were included…’ the text was changed (lines 137-139)

line 382: ‘… with do not perform these …’ The text was changed (lines 435-437)

Round 2

Reviewer 3 Report

The authors do a commendable job at incorporating suggestions from the previous round of review.